# Using early detection data to estimate the date of emergence of an epidemic outbreak

**Sofía Jijón**[1]*, **Peter Czuppon**[2], **François Blanquart**[3], **Florence Débarre**[1]*

**1** Institute of ecology and environmental sciences of Paris (iEES-Paris, UMR 7618), Sorbonne Université, CNRS, UPEC, IRD, INRAE, Paris, France, **2** Institute for Evolution and Biodiversity, University of Münster, Münster, Germany, **3** Center for Interdisciplinary Research in Biology, CNRS, Collège de France, PSL Research University, Paris, France

* sofia.jijon@gmail.com (SJ); florence.debarre@normalesup.org (FD)

## Abstract

While the first infection of an emerging disease is often unknown, information on early cases can be used to date it. In the context of the COVID-19 pandemic, previous studies have estimated dates of emergence (e.g., first human SARS-CoV-2 infection, emergence of the Alpha SARS-CoV-2 variant) using mainly genomic data. Another dating attempt used a stochastic population dynamics approach and the date of the first reported case. Here, we extend this approach to use a larger set of early reported cases to estimate the delay from first infection to the $N^{th}$ case. We first validate our framework by running our model on simulated data. We then apply our model using data on Alpha variant infections in the UK, dating the first Alpha infection at (median) August 21, 2020 (95% interpercentile range across retained simulations (IPR): July 23–September 5, 2020). Next, we apply our model to data on COVID-19 cases with symptom onset before mid-January 2020. We date the first SARS-CoV-2 infection in Wuhan at (median) November 28, 2019 (95% IPR: November 2–December 9, 2019). Our results fall within ranges previously estimated by studies relying on genomic data. Our population dynamics-based modelling framework is generic and flexible, and thus can be applied to estimate the starting time of outbreaks in contexts other than COVID-19.

**Data Availability Statement:** All data and codes needed for reproducibility of our results and the corresponding figures are available at DOI: 10.5281/zenodo.10657737.

## Author summary

While the first infection of an emerging disease is often unknown, information on early cases can be used to date it. In the context of the COVID-19 pandemic, previous studies have estimated dates of emergence of epidemic outbreaks (e.g., first human SARS-CoV-2 infection, emergence of the Alpha SARS-CoV-2 variant) using mainly genomic data. Another dating attempt used a population-level stochastic approach and the date of the first reported case. Here, we extend this generic and flexible approach to use a larger set of early reported cases to estimate the time elapsed between the first infection and the $N^{th}$ case. Our model dates the first Alpha infection at around August 21, 2020, and the first SARS-CoV-2 infection in Wuhan at around November 28, 2019. Our findings fall within ranges previously estimated by studies relying on genomic data.

**Funding:** SJ's postdoctoral fellowship was funded by a grant from the MODCOV19 platform of the National Institute of Mathematical Sciences and their Interactions (Insmi, CNRS) to FD. FD was funded by ANR-19-CE45-0009 (TheoGeneDrive). The funders had no role in study design, data collection and analysis, decision to publish, or preparation of the manuscript.

**Competing interests:** The authors have declared that no competing interests exist.

## Introduction

Dating the first infection of an emerging infectious disease is a step towards tracing the disease's origin and understanding early epidemic dynamics. Beyond the early transmission of a new pathogen, estimating the date of first infection is also of interest while studying the initiations of local clusters in naïve populations, such as when the pathogen is first introduced to a new location, but also when the pathogen evolves to distinct genotypes such as emerging variants of concern (VOCs).

Various attempts have been made to date the first human infections by SARS-CoV-2 that led to the COVID-19 pandemic (noting that earlier spillovers, leading to dead-ends, may have occurred), using case data and/or viral genomic data. Using a stochastic model for the epidemic spread coupled with genomic data allowing to trace transmission at the individual level, Pekar et al. [1] estimated that the first human infection took place between late October and early December 2019. This estimate was slightly later than a previous one by the same authors, who had previously found an emergence date between mid-October and mid-November 2019 [2]. Their revision notably included updating the dates of the first case reported [3], and better cleaning up genomic data to exclude sequencing errors [1]. Another, earlier, modeling study used case data only and dated the first COVID-19 case between early October and mid-November 2019, by adapting a technique used in conservation science for dating extinction using observation events [4]. The analysis had nevertheless been conducted on outdated case data [5]; re-running the analysis on updated case data had major effects on both the estimated date of the first infection and 95% confidence interval (see Fig A in S1 Text).

Other studies have focused on the introductions of SARS-CoV-2 to different countries. Some have used molecular clock analyses relying on genomic data to determine the time of most recent common ancestor (tMRCA; providing an upper bound for the time of the first infection, see Fig 1) of lineages introduced in a focal country. For instance, studies have been conducted using genomic data from France [6], the United States [7] and the United Kingdom (UK) [8]. Another study, based on case data only, used a stochastic non-Markovian approach relying on mortality data to estimate the date of SARS-CoV-2 introduction to France [9]; the first COVID-19 wave was estimated to have been initiated mid-January 2020 in France.

Finally, similar frameworks have been used to date the emergence of SARS-CoV-2 variants. One study focusing on the emergence of the 'EU1' SARS-CoV-2 variant (B.1.177) circulating among European countries during the summer of 2020 used genomic data and dated most introductions to June, 2020 [10]. Dating attempts have also been done for the 'Alpha' variant (B.1.1.7), whose date of emergence was estimated at early August 2020 using a stochastic, non-

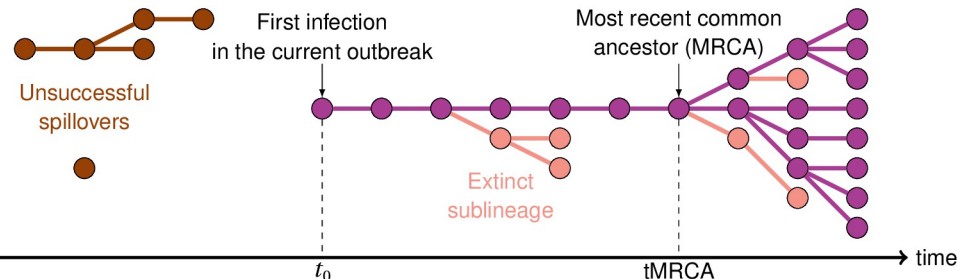

**Fig 1. Schematic of early transmission dynamics.** Each dot corresponds to an infected individual, and the link between dots represent pathogen transmission. Only a fraction of all infections will be detected as cases. We aim to estimate $t_0$, the time of the first infection which is at the origin of the ongoing outbreak. Genomic methods can date a most recent common ancestor (MRCA).

Markovian approach relying on the date of the first observed case [11], and whose tMRCA was estimated at late August 2020 [12].

Genomic analyses on their own are usually insufficient to estimate the time of the first infections: the tMRCA does not necessarily approximate the emergence date [13], which can have taken place earlier (see Fig 1). Infection times occurring earlier than the tMRCA may be estimated thanks to other mathematical models using population dynamics methodologies. Moreover, modeling studies have helped unveiling other unobserved indicators during the early stages of epidemics, such as the epidemic size at the time of first detection [11] or at the time of the first reported death [14]. In particular, because infection numbers are low, stochastic approaches are key to studying early dynamics and their long-term effects on the epidemic outbreak [15–19]. Hence, methodological developments of stochastic models to study the early stages of infectious diseases remain of great interest in the field of mathematical epidemiology; e.g. [20]. Finally, we think that there is value in approaching an inference question via different methods (especially on potentially contentious topics), in order to obtain methodologically independent confirmations of key estimates.

The main objective of our study is the estimation of the date of the first infection leading to a sustained epidemic (hereafter referred to as the date of epidemic/outbreak emergence), using available data on the first $N$ detected and reported cases. To this end, we build a stochastic model and design a simulation framework extending our previous work that only used information on the first detected case [11]. Our rationale for extending the analysis to more cases is two-fold. First and foremost, including more cases can make the analysis more robust to changes in the underlying case data. Second, a new methodology using more than just the first case will also give access to important information such as the proportion of detected cases among infections. Our simulations construct a transmission tree, and identify detected cases among the infections. The comparison of simulation outputs and available case data then allow the estimation of the date of emergence of the outbreak, as well as other key epidemiological pieces of information such as the proportion of the epidemic that remains undetected. We apply the framework to two examples: the emergence of the Alpha variant in 2020 in the UK, and the first human SARS-CoV-2 infection in Wuhan in late 2019.

## Results

### Modeling the early dynamics of an epidemic outbreak

**Model description.**   We develop a stochastic epidemic model that estimates the time elapsed between the first infection and $N$ reported cases (i.e., infections that were detected and reported), using available data and estimates on key epidemiological parameters. We model infectious disease transmission with a general branching process starting from a single infectious individual. This implies that times from infection to transmission events are not exponentially distributed as assumed by ordinary differential equation models. We then model the detection of infected individuals, which constitutes the modeled time series of cases. Both infection and detection processes follow distributions with known fixed parameters.

Importantly, we assume fixed parameters as estimated by previous studies (in particular, the detection probability), but we later examine the robustness of our findings to these exact values. We consider the time-series of infections and detections up to the day of occurrence of the $N^{\text{th}}$ case. We calibrate our model to reproduce the observed epidemic, using available data on disease cases. For more details on the model, we refer to the Methods section.

Our model generates a time series of cases, from which we deduce the delay between emergence (first infection) and $N^{\text{th}}$ case. By first infection, we refer to successful epidemic outbreaks only; that is, we do not account for the first infections that may have led to epidemics that

**Table 1. Estimates of the date of emergence and other epidemiological indicators resulting from the calibrated model.** Here we summarize the estimates obtained from the numerical application of our model to two epidemiological contexts: the Alpha variant infections in UK, and the first COVID-19 cases reported in Wuhan. Median and 95% interpercentile (IPR) ranges across retained simulations are shown, unless stated otherwise. The estimated time elapsed between the first infection to the $N^{th}$ observed case yields the estimated emergence date. The earliest date corresponds to the lower bound of the emergence dates distribution. In addition, we estimate the epidemic size at the date of detection of the $N^{th}$ case. The proportion of detected infections and mean secondary cases are retrieved for comparison with the input epidemic parameters.

| Epidemiological indicator | Alpha (UK, 2020) | COVID-19 (Wuhan, 2019) |
|---|---|---|
| Number of days from $1^{st}$ infection to $N^{th}$ case* | 82 (67–111) | 52 (41–78) |
| Date of first infection | Aug 21 (Jul 23–Sep 5), 2020 | Nov 28 (Nov 2–Dec 9), 2019 |
| Date of earliest infection | Jun 13, 2020 | Sep 26, 2019 |
| Epidemic size at day of infection of the $N^{th}$ case | ∼ 90 800 (80 500–102 000) | ∼ 63 400 (57 300–69 900) |
| Proportion of detected infections | 0.48% (0.43%–0.52%) | 5.31% (5.06%–5.57%) |

* $N$ = 406 on November 11, 2020 for the Alpha application, and $N$ = 3072 on January 19, 2019 for the COVID-19 application.

went extinct. In addition, by keeping track of the whole epidemic (i.e, the time series of infections from which we construct the time series of detections), we retrieve, for instance, the number of secondary cases produced by each infectious individual and the time series of infections that are detected at a later time. We use the latter to compute the epidemic size at the time where the $N^{th}$ case is infected and deduce the proportion of infections that are detected (i.e., cases) within the study period (i.e., up to the day of detection of the $N^{th}$ case). This proportion is impacted by detection delays and stochasticity and thus, it is not straightforwardly obtained from the probability of detection considered in the simulations.

We run as many numerical simulations as needed to obtain 5 000 successful epidemics, i.e., epidemics that were sustained after a predetermined period of time and that verified the calibration conditions imposed by the data from a certain epidemiological context (details in the Methods section). We apply our model to two epidemiological contexts: the emergence of the 'Alpha' variant in the UK and the emergence of SARS-CoV-2 in Wuhan. Both applications and the corresponding results are described in more detail below and summarized in Table 1.

**Testing the model on simulated data.** To check the validity of our framework, we ran our model to generate epidemics and create case data from them. With these simulated data, the actual date of the first infection is known. We then applied our framework to generate estimates of the date of first infection, and compared it to the actual dates. We confirmed that our framework could recover the date of the first infection, the median being within a 2–3 day delay (see Figs B and C in S1 Text for details). Median estimates of the date of the first infection were not much affected by changes in the number of considered cases, $N$. Confidence intervals, and therefore the range of potential dates, are larger as we include more cases, in particular because our conditions to retain a simulation (see Methods) are less stringent on the first cases, as the total number of cases increases. We also note that the estimated date of first infection is more likely to be earlier rather than later than the simulated date of first infection, i.e., the delay between the first infection and first detected case is more likely to be overestimated than underestimated.

## Estimating the date of the first infection with the Alpha variant in the UK

Moving on to real case data, we first ran our model to estimate the date of emergence of the Alpha variant in the UK, which was the main result of the numerical applications presented in

[11]. We applied our model to a dataset of $N$ = 406 samples carrying the Alpha variant collected and sequenced between September 20 and November 11, 2020 [21] (cf. Fig D in S1 Text). Table 2 summarizes the parameter values used in our simulations. The 5 000 simulated epidemics that we analyse below result from model calibration (i.e., epidemics arising from a single infectious individual and verifying the calibration constraints; details in the Methods section), and represent $\sim$36% of all simulations run with the input parameters. The cumulative cases of the accepted epidemics are depicted in Fig E in S1 Text.

Hereafter, we summarize our results using median values and 95% interpercentile ranges (95%IPR; values between the $2.5^{\text{th}}$ and the $97.5^{\text{th}}$ percentiles) from the distributions of the different epidemiological indicators obtained from the 5 000 simulated epidemics, similar to an approximation of the posterior distribution obtained in an Approximate Bayesian Computation framework (see Methods). We also use the minimum values from those distributions to unveil the earliest infection occurrences among our simulated epidemics (that is, the earliest dates possible for the first infection to occur).

We estimated the number of days between the $1^{\text{st}}$ infection and the $N^{\text{th}}$ case at 82 (95%IPR: 67–111), dating the emergence of the Alpha variant in the UK at August 21 (95%IPR: July 23–September 5), 2020, and not earlier than June 13, 2020. Alpha transmissions were ongoing about 30 days (95%IPR: 10–60) before the date at which the first known case was sampled and sequenced. Table A in S1 Text provides other calibration metrics like the delay between the $1^{\text{st}}$ and $N^{\text{th}}$ sequenced samples in the simulations. Fig 2 depicts our estimates of the date of emergence along with the epidemic curve (i.e., the daily number of sequenced samples; by sampling date), for context, as well as previous estimates, for comparison. In particular, we ran an updated version of the model presented by Czuppon et al. in [11] (the distribution of the number of secondary cases is negative-binomial instead of Poisson previously, and its mean, $R$, is now equal to 1.9 instead of 1.5 in [11]). We also compare our results to tMRCA estimates by Hill et al. [12] (personal communication of the distributions). Our median estimates for the emergence date fall within a very close, slightly narrower range than that found by running an updated version of [11], while falling $\sim$ 1 week earlier than the estimated tMRCA [12]. These comparisons are summarized in Table B in S1 Text.

**Table 2. Input parameters.** Dates for the first and $N^{\text{th}}$ observed cases correspond to the dates in the data set. A case in the context of the Alpha variant in the UK is defined as a sequenced sample, whereas a case in the context of COVID-19 in Wuhan is defined as a confirmed, symptomatic case. The total numbers of observed cases correspond to the size of the data set used to inform the model. The key epidemiological parameters are obtained from available literature (references in brackets, next to the parameter values).

| Parameter | Symbol | Alpha | SARS-CoV-2 |
|---|---|---|---|
| Total number of observed cases | $N$ | 406 | 3072 |
| Date of first reported case | $d_1^{\text{obs}}$ | Sep 20, 2020 | Dec 10, 2019 |
| Date of $N^{\text{th}}$ observed case | $d_{K'}^{\text{obs}}$ | Nov 11, 2020 | Jan 19, 2020 |
| Expected number of secondary cases | $R$ | 1.90 [12] | 2.50 [37] |
|  | $\kappa$ | 0.57 [11] | 0.10 [38] |
| Secondary infection generation time | $\omega_t$ | 0.83 [39][†] | 0.83 [39] |
|  | $\theta_t$ | 6.60 [39][†] | 6.60 [39] |
| Probability of case detection | $p_{\text{detect}}$ | 0.01 [40, 41][†] | 0.15 [37] |
| Time from infection to detection | $\omega_\tau$ | 0.58 [11] | 1.04 [42] |
|  | $\theta_\tau$ | 12.0 [11] | 6.25 [42] |
| Tolerance on the daily number of cases* | $\delta_{\text{tol}}^Y$ | 0.3 | 0.3 |

\* Used to select the simulations that resemble the observed data; cf. the Methods section.

[†] We use the parameters from the studies cited in [11].

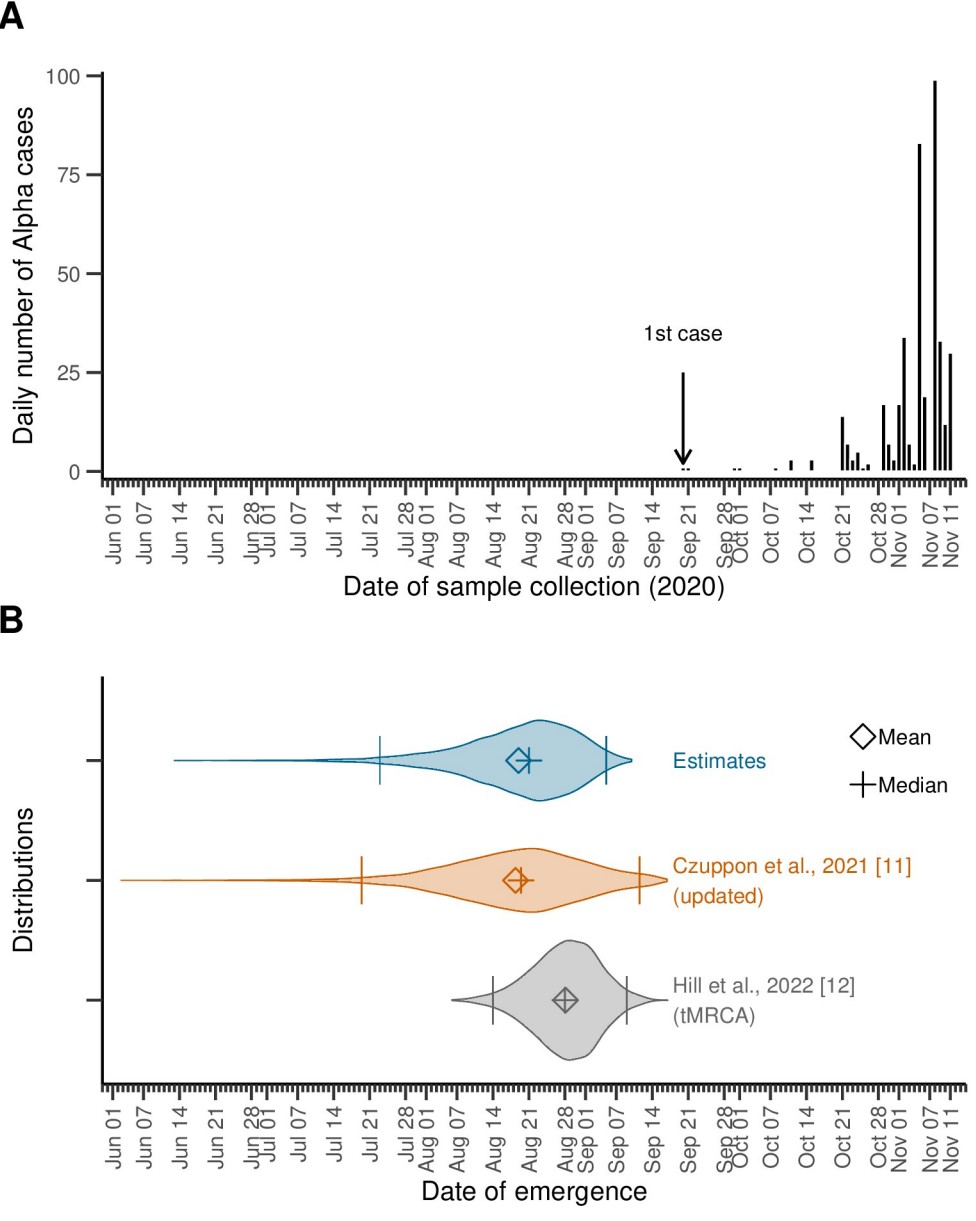

**Fig 2. Estimates of the date of emergence of the Alpha variant in the United Kingdom (UK). (A)** The epidemic curve corresponding to the data used to calibrate the model, for context. A total of $N$ = 406 samples carrying the Alpha variant were collected between September 20 and November 11, 2020 [21]. **(B)** Violin plots for the distributions of the date of emergence. We estimated the emergence of the Alpha variant in the UK at August 21 (95%IPR: Jul 23–Sep 5), 2020 (top, blue; upper and lower bound of the 95%IPR depicted by bars). For comparison, we also show the distributions of the estimates from an updated version of the model developed in [11]—which relies on a single observation on September 20—where we set $R$ = 1.9 and a negative-binomial distribution for the number of the secondary cases (middle, orange). The distribution for the estimated time of most recent ancestor (tMRCA) [12] is also shown (bottom, gray).

We further estimated the epidemic size at the date of infection of the $N^{\text{th}}$ case at about 90 800 (95%IPR: 80 500–102 000). The simulated detected cases thus represent a proportion of about 0.48% (95%IPR: 0.43%–0.52%) of the total number of infections. Note that in this section, a case is an infected individual who underwent a PCR test and whose sample was

sequenced and contained the Alpha variant, which accounts for the low probability of detection. Our results are summarized in Table 1. Note that when $N = 1$, we recover results similar to [11], once parameters are updated to match, cf. Fig F in S1 Text.

## Estimating the date of emergence of SARS-CoV-2 in Wuhan

Next, we applied our model to the dataset of the early cases of COVID-19 considered in [1] (personal communication). A total of 3 072 confirmed COVID-19 cases were reported to have had symptoms onset between December 10, 2019 and January 19, 2020, the day before the first public declaration of human-to-human transmission, shortly before the first lockdown interventions [22] (cf. Fig G in S1 Text). We parameterized our model using estimates from the literature; they are listed in Table 2. The 5 000 selected simulations represent $\sim 14\%$ of all runs (cf. Table A in S1 Text) and are depicted in Fig H in S1 Text.

Our simulations yield an estimated median number of days between the $1^{\text{st}}$ SARS-CoV-2 infection to the $N^{\text{th}}$ symptomatic COVID-19 case recorded of 52 (95%IPR: 41–78) days, dating the emergence (i.e., the first sustained human infection) of SARS-CoV-2 to November 28 (95%IPR: November 2–December 9), 2019, and not earlier than September 26, 2019. This also implies that the epidemic remained completely undetected (i.e., no detected infections) for about 9 days (95%IPR 3–21). These findings are depicted in Fig 3, along with the observed epidemic curve (i.e., COVID-19 cases dataset) as well as previously published estimates of the date of emergence of the COVID-19 pandemic [1] (personal communication of the distributions), for comparison. Our estimates of the date of first infection are consistent with those previously found in [1], but yield a lower bound closer to the date of first case detection (cf. Table B in S1 Text). We also compare our results to those obtained through other more computationally expensive methods (namely, Approximate Bayesian Computation approaches) to illustrate the benefit of using our model.

We further estimate the median number of infections on the day of infection of the $N^{\text{th}}$ case at about 63 400 (95%IPR: 57 300–69 900), which results in a median proportion of detected infections of 5.31% (95%IPR: 5.06%–5.57%). Our results are summarized in Table 1.

## Sensitivity analyses

Here, we evaluate the impact of uncertainty around the main model parameters on the results, by running our simulations while varying the input values of the expected number of secondary cases ($R$), the overdispersion parameter ($\kappa$), the probability of detecting an infection ($p_{\text{detect}}$) and the number of days elapsed between infection and detection, via the distribution parameter $\theta_\tau$. We find that, as expected, increasing $p_{\text{detect}}$, $R$ or decreasing $\kappa$, $\theta_\tau$ results in a reduction in the number of days between the first infection and the $N^{\text{th}}$ case, meaning that the epidemic emerges later in time; cf. Fig 4A–4H, and Table C in S1 Text. Median estimates and 95%IPR are summarized in Table B in S1 Text.

We further evaluate the impact of the criteria chosen for accepting the simulated epidemics on our results (cf. Methods section for details). We run our simulations while varying the tolerance for the difference between the simulated and the observed daily number of infections, $\delta^Y_{\text{tol}}$, for both applications. Our estimates are robust with respect to these variations (cf. Fig 4I and 4J).

**Uncertainty around very early case declaration.**   We ran additional sensitivity analyses for the application to the emergence of the COVID-19 cases in Wuhan, to evaluate the impact of the uncertainty around very early case declaration, by applying our model to different datasets; cf. Section 5.1.2 in S1 Text. We first used a shorter dataset, with data on COVID-19 cases with symptoms onset only up to December 31, 2019, the day of the first public declaration of a

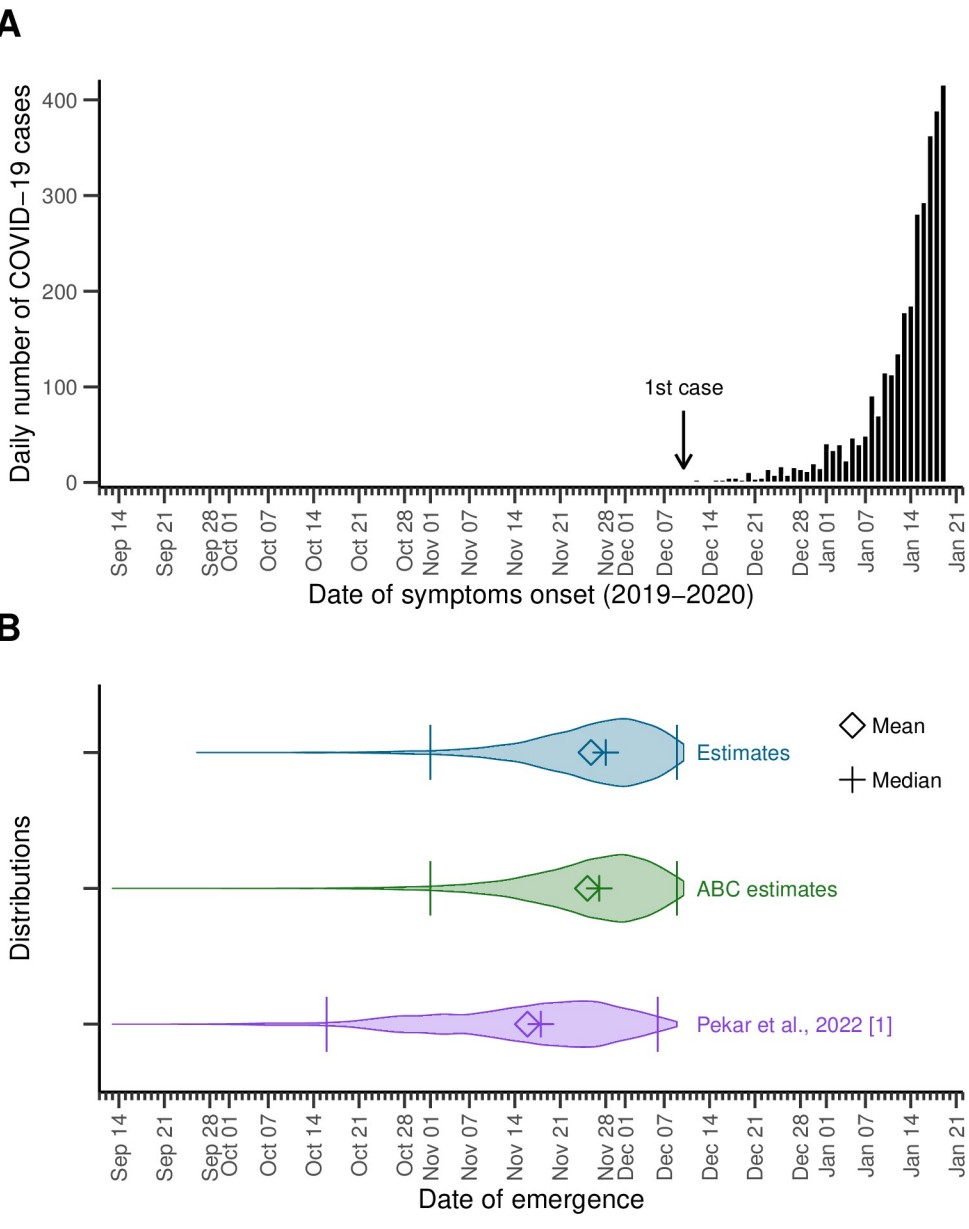

**Fig 3. Estimates of the emergence of SARS-CoV-2 in Wuhan. (A)** Observed epidemic curve, for context. A total of *N* = 3 072 COVID-19 cases with symptom onset between December 10, 2019 and January 19, 2020, the day before the first public statement on human-to-human transmission. NB. The scale of the figure makes the first case and other bins corresponding to sole cases hardly visible; we refer the reader to Fig G in S1 Text for a larger picture of the epidemic curve. **(B)** Violin plots for the distributions of the date of emergence. We estimated the median date of the first SARS-CoV-2 infection (i.e., emergence) at November 28 (95%IPR: November 2–December 9), 2019 (top, blue; upper and lower bound of the 95%IPR depicted by bars). For comparison, the distribution for the estimates obtained implementing more classical approximate bayesian computation (middle, green), and estimates from [1] (bottom, purple) are also shown.

cluster of pneumonia of unknown etiology [22] (*N* = 169). We also used an outdated, later corrected dataset published by the WHO in 2020 [23] (*N* = 202) where the first case reported symptoms onset was on December 2, 2019. We found that using the shorter dataset dates the epidemic emergence 2–3 days earlier than our main results, and selects simulations with a

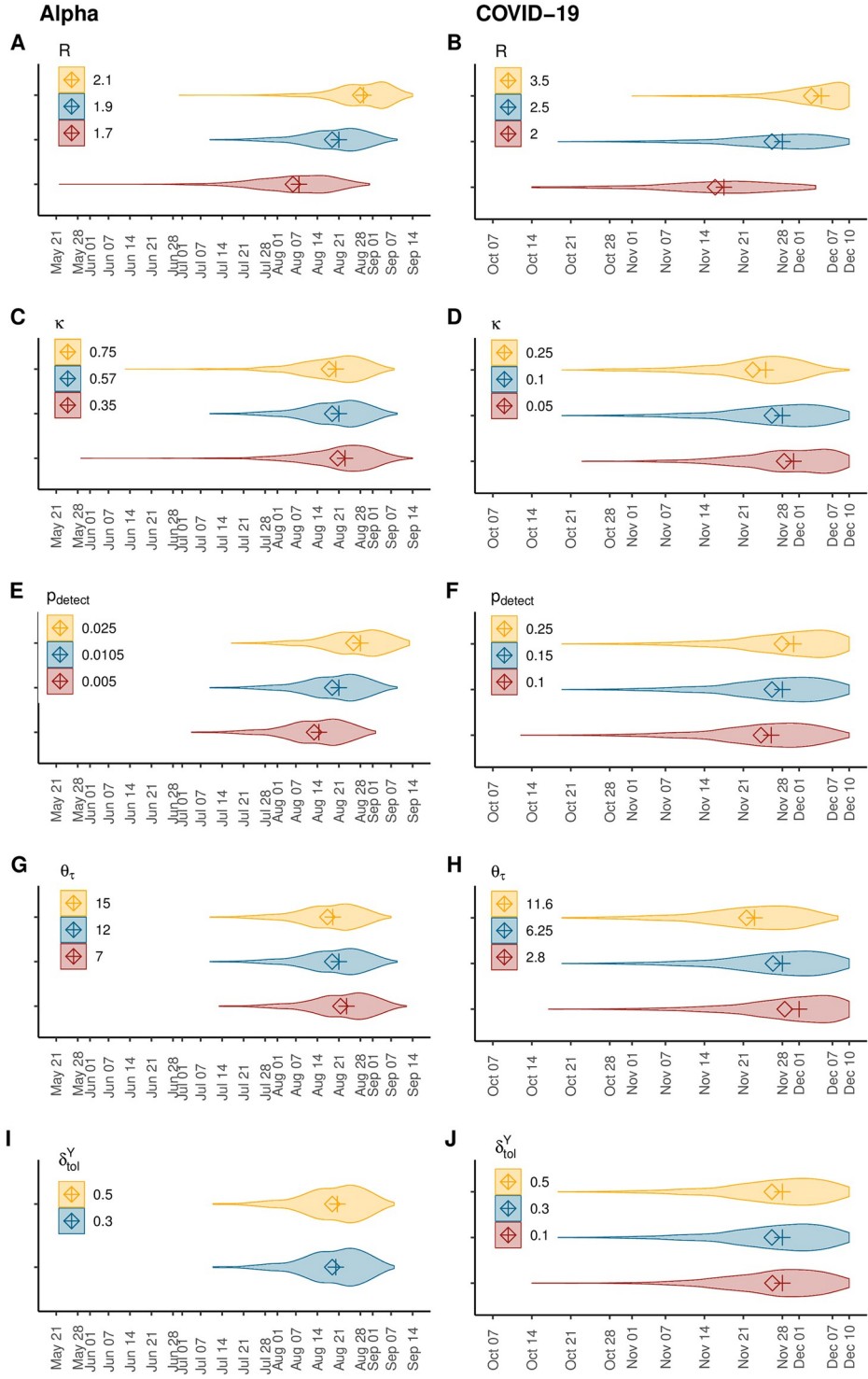

**Fig 4. Sensitivity analyses.** Distributions for the date of emergence of the Alpha variant in the UK (left) and the COVID-19 epidemic in Wuhan (right) obtained from running our simulations setting different input values for key model parameters: the reproduction number, $R$ (panels A and B), the overdispersion parameter, $\kappa$, (panels C and D), the probability of detection, $p_{\text{detect}}$ (panels E and F), the time elapsed between infection and detection, by varying $\theta_\tau$ (panels G and H) and the tolerance for the difference between the simulated and the observed daily number of infections, $\delta_{\text{tol}}^Y$ (panels I and J). The absence of a violin plot for $\delta_{\text{tol}}^Y = 0.1$ in panel G results from the absence of selected simulations in > 3 million runs. Median values are depicted by crosses and mean values by diamonds. Baseline parameterization of the model is depicted in blue (middle).

slightly higher mean number of secondary infections (c.f. Table 1 and Table D in S1 Text), which may reflect a change in the population behavior following the first public announcement. However, using the outdated WHO 2020 dataset [23] dates the epidemic emergence about a week prior to our main results, while the mean number of secondary infections remained about the same, which reflects the (relatively limited) impact of the date of first detection on the results. These results are depicted in Fig I and summarized in Table D in S1 Text.

## Discussion

We estimate the date of emergence of an epidemic outbreak, defined as the date of the first infection leading to a sustained transmission chain, relying on estimates of key epidemiological parameters as well as available data on the first $N$ observed cases. With our population-dynamics approach, we recover estimates very close to those of previous studies [1, 12], which were obtained using information from whole genome sequences. Our results constitute methodologically independent confirmations for the previously published estimates.

Our model was conceived as an extension of the numerical application presented in [11]. This methodology relies on a general branching process to model disease transmission and detection. Our model is informed by available data on the first $N$ observed cases (unlike [11], who used the date of first detection only). We further account for super-spreading using a negative-binomial distribution for the generation of secondary infections. This assumption may reduce the time elapsed between the first infection and the first detection or increase its variance, in comparison to ignoring super-spreading by considering other distributions (e.g. Poisson) [11] or by ignoring individual heterogeneity in infectiousness [9], as well as by using deterministic approaches [9].

After validating our approach with simulated data, we first studied the emergence of the Alpha variant. Our results suggest that the Alpha variant emerged in the UK around August 21, 2020 and not earlier than June 13, 2020. Our results fall indeed within the same ranges as those of the approach presented in [11] when updated to match our parameterization, and fall shortly earlier than previous tMRCA estimates of the Alpha variant [12]. This result makes sense: the tMRCA does not necessarily yield the date of first infection, which may have occurred before the most recent common ancestor.

Next, we apply our model to data on the early COVID-19 cases in Wuhan, estimating the date of the first SARS-CoV-2 infection around November 28 (95%IPR: Nov 2–Dec 9), 2019, and not earlier than September 26, 2019. These ranges also fall remarkably close to—slightly later than—previously published estimates [1]. To the best of our knowledge, the novelty of our findings rests on using exclusively a population-dynamics approach, unlike previous studies aiming to date the emergence of the COVID-19 epidemic. The median estimate in Pekar et al. [1] (which is a first infection, and not a tMRCA) falls $\sim 8$ days before ours, which can be explained by the fact that our approach ignores genomic information. Namely, with our approach, having two cases with the exact same infecting virus, or two cases with viruses two mutations away, are treated the same way, as our approach only uses case numbers. Using genomic information would however yield an earlier date of emergence if the infecting viruses are genetically more distant. Our findings are thus also compatible with a previous study that found no evidence of widespread transmission in Wuhan before December 2019, using serological data [24]. Hence, in accordance to previous discussions [1, 2], our results suggest that widespread SARS-CoV-2 circulation (and even more so, international spread) earlier than the end of 2019 is unlikely. Assuming an origin of the pandemic in China [25], claims of large early (i.e., before January 2020) circulation outside of China (e.g. [26] and references therein)

would be therefore extraordinary, and require extraordinary evidence, excluding potential false positives by setting appropriate controls. Our study differs from [1, 2] in that we use population-level rather than genomic data, and does not intend to replace nor outperform other approaches. Rather, our results provide further support for these previous estimates relying on different methods.

Quantifying the time that emerging epidemics remain undetected before detecting the first cases is particularly important in the context of emergent pathogens such as SARS-CoV-2, where very early cases may remain unidentified, especially if a high proportion of the infections are asymptomatic [27] (N.B.: most COVID-19 cases with symptoms onset up to December 31, 2019 were declared retrospectively [28]). There is also evidence that SARS-CoV-2 may have been introduced in other countries for some time before the first reported cases in these countries [6–8, 29]. Since the first known cases are likely not to be the first human infections, there is a need for methods estimating the time of the first infection. The value of such work is essentially historical, but also helps understand how long an epidemic may have gone undetected, and design early warning systems accordingly. For instance, our approach shows that early SARS-CoV-2 infections of the transmission chain that was first detected at Wuhan's Huanan market [25] remained undetected for between about a week and three weeks before the first case of symptomatic COVID-19 was observed and reported.

The impact of uncertainty on our results is assessed by varying the main transmission and detection parameters, as well as the rejection criteria for the simulated epidemics. These sensitivity analyses show the robustness of our approach regarding the choice of the rejection criteria. On the other hand, we find that the main model parameters (expected number of secondary infections, $R$, and probability of detection, $p_{\text{detect}}$) have a greater impact on the model outcomes: as expected, both higher transmissibility and higher detection shorten the time between emergence and $N$ detected infections. This variation in the results is particularly true for the application on the Alpha variant data, probably due to the notably smaller case dataset we use.

Our study has several limitations. First, our results depend heavily on input data, while access to good quality data on the early stages of an epidemic outbreak may be challenging. Datasets may be scarce, they may face reporting delays, early cases may be detected retrospectively and detection protocols may change. Early Wuhan COVID-19 cases with symptoms onset before December 30 (the day of issue of the emergency notice from the Wuhan Municipal Health Commission) [22, 30] were diagnosed clinically before tests for SARS-CoV-2 infection were available [3]. Second, our model requires early estimates of the distributions for key epidemiological indicators such as the mean number of secondary infections, the secondary infection generation time, the probability of case detection and the incubation period, which depend themselves on the quality of early observed data and may not be available for new emerging infectious diseases. In particular, it can be challenging to estimate the probability of case detection (or ascertainment rate) during early stages of an emergent infectious diseases, and its value may vary between countries [31]. Third, our methods rely on the hypothesis that by the time an infection is detected, a sustained epidemic is ongoing, and all cases contained in the data set belong to the same transmission tree. That is, our model does not deal with earlier transmission trees that have gone extinct. We model epidemic spread starting from a single infectious individual, thus neglecting scenarios of multiple introductions. This impedes, for instance, the application of our model to contexts such as SARS-CoV-2 importation to France [6]. This may also be a limitation in the context of epidemics emerging from multiple spillover events, such as has been concluded by [1] relying on data on the early SARS-CoV-2 lineages. Fourth, the simulations of our model do not allow to consider time-dependent parameters, but a more classic Approximate Bayesian Computation approach will. Hence, we are

constrained to use data on relatively short periods of time to ensure that epidemiological parameters remain nearly constant over the study period. This may not reflect early epidemic dynamics, where public outbreak alerts may provoke, on the one hand, an increase in testing efforts and thus, rapid changes in the probability of detection and, on the other hand, changes in individual behaviors that may impact the probability of disease transmission and thus, the expected number of secondary infections.

The numerous, fast and free availability of genomic data for the COVID-19 epidemic is unprecedented. Here, we built our model in a parsimonious, generic and flexible manner intended to be applied to contexts other than COVID-19, provided that key epidemiological parameters are known and transmission chains arise from a single infectious individual. Further developments of our model need to include genomic data on top of case data, as it is likely that sequencing will remain as intensive for other infectious diseases as it has been for COVID-19.

In conclusion, our study contributes to the literature with estimates of the date of first infection of early COVID-19 cases and infections with the SARS-CoV-2 Alpha variant. Our results fall within ranges previously found by studies relying on genomic data, thus offering weight to these estimates using different methodologies.

## Methods

### Model

We extend the methods presented in the numerical applications of [11], where the time of emergence was estimated from data on the first reported case only, using a stochastic population-dynamics approach. We model the early stages of the epidemic and use dates of the first $N$ observed cases to estimate the date of emergence. We use the term '*case*' to refer to infections that are ascertained and reported: time series of cases may thus correspond to one of the following types of time series: infection detection, sample sequences, symptoms onset declarations, etc. We use the term '*probability of detection*' for the probability of such ascertainment to occur.

Our model is defined by a non-Markovian branching process to model the transmission of an infectious disease, starting from a single infectious individual in a fully susceptible population [9, 11]. Since we study early epidemic dynamics, i.e., for a relatively short period of time, all distribution parameters are assumed to be constant during the modeled time period. At any time $t$, infected individuals may transmit the disease. We account for large numbers of transmissions generated by few individuals, '*super-spreading*', by assuming that the number of secondary cases follows a negative binomial distribution,

$$NegBinom\left(\text{number of failures} = \kappa, \text{probability of transmission} = \frac{\kappa}{\kappa + R}\right), \quad (1)$$

where $R$ denotes the expected number of secondary infections (i.e., the effective reproduction number). Then, we model the ascertainment of infections by drawing the number of detected cases among the secondary infections from a Binomial distribution

$$Binom\left(\text{number of trials} = I(d_k), \text{ probability of success} = p_{\text{detect}}\right), \quad (2)$$

where $I(d_k)$ is the number of incident infections at day $d_k$, with $k = 1, 2, \ldots$, and $p_{\text{detect}}$ is the probability of infection detection. The generation time of each new infection, $\{t_i\}_{i=1,2,\ldots}$, as well as the time from infection to detection of a case are drawn from a Gamma distribution

$$Gamma\left(\text{shape} = \omega_x, \text{ scale} = \theta_x\right), \quad (3)$$

with $x \in \{t, \tau\}$, respectively, where we use $\{\tau_j\}_{j=1,2,\ldots}$ to denote the time series of observed cases. N.B. we assume that $t$ and $\tau$ are independent; that is, we do not model the generation time depending on the infectious individuals' own time elapsed between infection and detection. Note that the finite number of secondary infections per individual together with the generation time distribution mean that an individual is able to transmit the pathogen for a limited period of time only.

## Estimation of the date of first infection

In our simulations, we discretize time by $\Delta t = 0.1$ days. Note that times to infection and detection are drawn from their corresponding distributions and thus kept on a continuous scale. We then aggregate epidemiological indicators such as the per-day number of new infections and cases (denoted by $d_k$, where $k = 1, 2, \ldots$), since this is the time scale at which most data are presented. We run stochastic simulations forward in time, from $t = t_0$, the time of occurrence of the first (undetected) infection (i.e., $I(t_0) = 1$), until the end of the day of detection of the $N^{\text{th}}$ case, $d_K$, which is determined by $d_K \leq \tau_N < d_{K+1}$. Note that while the $N^{\text{th}}$ case is a stopping criterion of our model, our analyses still deal with all $M \geq N$ cases occurring up to the end of day $d_K$: the number of infections detected on day $k$ is defined by

$$Y_k \equiv \sum_{j=1}^{M} \mathbb{1}_{d_k \leq \tau_j < d_{k+1}}, \qquad k = 1, 2, \ldots, K. \tag{4}$$

A depiction of our infectious disease transmission and detection model is shown in Fig 5.

The goal is to estimate the date of the first infection. To this end, we follow a strategy that is similar to an Approximate Bayesian Computation (ABC) algorithm, but meant to be computationally more efficient. In ABC, one would define a (typically uniform) prior distribution for the date of the first infection (i.e., date of emergence), randomly draw emergence dates from this distribution, simulate stochastic epidemics and compute a distance between the data (the observed time series of cases) and each simulation. One simple algorithm (*rejection* algorithm) consists in accepting the fraction of trajectories that are closest to the data. The distribution of emergence dates from the accepted simulations then approximates the posterior distribution of the date of first infection. For comparison, we have implemented this computationally intensive ABC strategy for the COVID-19 (Wuhan) data set. We simulated 10 000 established epidemics, i.e. the simulated epidemic reached at least 1 000 infections, for each possible date of first infection, ranging from September 2 till December 10, 2019 (the end date corresponding to condition (C1) below). Simulations were stopped when the simulation reached January 19, 2020, the date of the 3 072$^{\text{th}}$ detected case in the empirical data set. Simulated epidemics were then rejected based on condition (C2) below to obtain the approximate posterior distribution of the first infection dates. Using this approach, we retained 3.5% of all established epidemic simulations, which is considerably less than the 14% retained simulations with our alternative approach, which we outline now.

Our inference method is an alternative to this computationally intensive ABC algorithm and yields an approximate posterior distribution of the date of emergence. In a nutshell, this method makes use of the fact that parameters are constant in time, and that dates are therefore arbitrary; more than absolute dates, what matters in the end is the time *interval* between the first infection and the $N^{\text{th}}$ case. We can therefore run simulations without a reference to absolute time; we define absolute time in a second step only, by aligning the day at which the $N^{\text{th}}$ case is attained (if ever) in the simulation with the day at which the $N^{\text{th}}$ case is attained in the data. Thanks to this property, we can sample one accepted trajectory after another and have more precise control on the number of accepted simulations that are underlying the

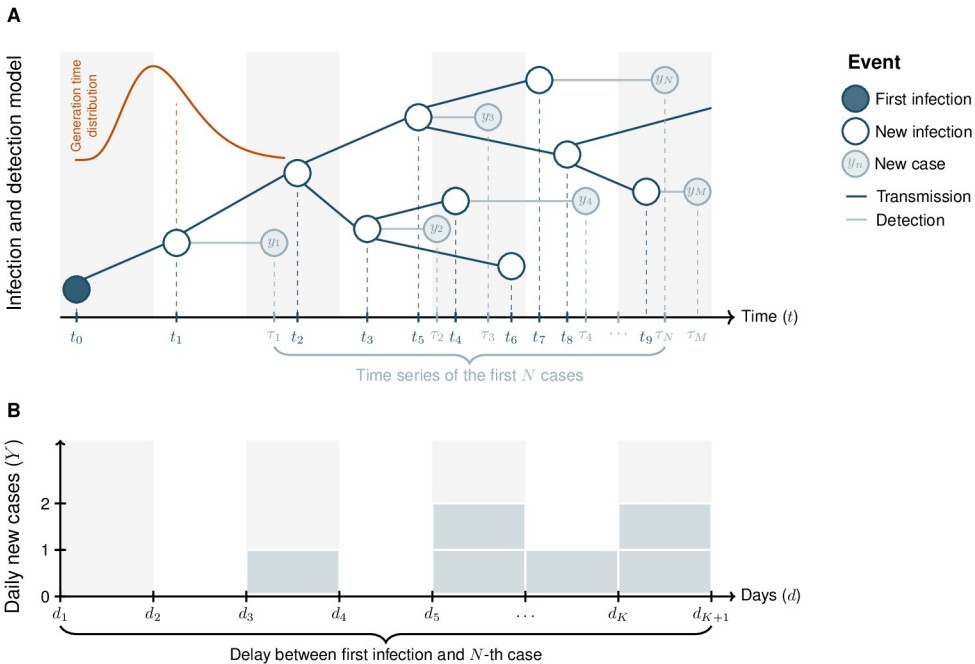

**Fig 5. Model diagram. (A)** We model infectious disease transmission (dark blue elements) starting from a single infectious individual (full dark blue dot), using a general branching process. The generation time of secondary infections, $\{t_{i+1} - t_i\}_{i=0,1,...}$, follows a Gamma distribution (shown in orange, above the first transmission event). In addition, we model the detection of infected individuals (light blue elements), which yields the time series of observed cases, $\{\tau_j\}_{j=1}^M$, with $M \geq N$. The number of cases are aggregated daily. Days are denoted by $\{d_k\}_{k=1}^K$ and depicted by alternating gray and white bands. Our algorithm stops the day at which the $N^{\text{th}}$ case is observed, $d_K$, but our analyses deal with the set of all cases detected on day $d_K$, $\{y_j\}_{j=1}^M$, where $y_j$ denotes the $j^{\text{th}}$ detected infection (i.e., the $j^{\text{th}}$ case). **(B)** Resulting epidemic curve (cases per day). Our model is calibrated so that the simulated epidemic curve, $\{Y_k\}_{k=1}^K$, reproduces the observed number of cases per day. The main outcome of our model is the number of days elapsed between the first infection and the $N^{\text{th}}$ observed case, $d_K$. NB. The time scale in the figure is not representative of our simulations: infection and detection delays in the simulations usually span multiple days.

approximate posterior distribution. In contrast, with a classical ABC approach there are a lot more of rejected simulations (and wasted computation time). Our algorithm is an order of magnitude faster than classical ABC because all our simulations are constrained to match exactly the observed date of the $N^{\text{th}}$ case.

The conditions to accept or reject a simulation are the following. Simulated epidemics ('sim') are calibrated to reproduce the observed ('obs') epidemic, via two conditions.

First, we require that the first simulated infection predates the first observed case:

$$t_1^{\text{sim}} \leq \tau_1^{\text{obs}}. \tag{C1}$$

Second, we accept simulations in which an epidemic occurs with enough cases, and we require the epidemic curve to resemble the whole time series of case data. More specifically, the daily number of cases of the simulated epidemic is required to resemble (under a certain tolerance $\delta_{\text{tol}}^Y$) the observed behavior:

$$\max_k \left| \sum_{j=1}^k Y^{\text{obs}}(d_j) - \sum_{j=1}^k Y^{\text{sim}}(d_j) \right| \leq \delta_{\text{tol}}^Y N, \qquad k = K, K-1, K-2, \ldots, \tag{C2}$$

where $\sum_{j=1}^{k} Y(d_j)$ denotes the cumulative number of cases at day $d_k$. The daily case count in the accepted simulations thus depends on $N$, i.e. on the epidemiological context.

This procedure is repeated until we simulated 5 000 accepted epidemics. To compute the dates of emergence from these simulations, we take the final date of the observed case data and subtract the duration of each simulated epidemic, which produces the posterior distribution of the date of emergence.

For more details on the numerical application of the model, please refer to the pseudo-algorithm in S1 Text. The simulations were run in Julia [32] version 1.8, and the results figures were generated in R [33] version 4.1.2, using the `ggplot2` package [34], version 3.3.6. All data and codes needed for reproducibility of our results and the corresponding figures are available at https://doi.org/10.5281/zenodo.10657737.

## Applications

**Alpha variant in the UK.** The first application concerns the early stages of the spread of the Alpha SARS-CoV-2 variant in the UK. Note that the case $N = 1$ thus serves to confirm whether our extended version of the model presented in [11] recovers its results.

The first reported sequence of the SARS-CoV-2 Alpha variant of concern was collected on September 20, 2020 [35]. Here, we define a case as a sequenced sample carrying the Alpha variant, and we define the probability of a case detection as the probability of sampling *and* sequencing such variant. The parameterization of our model is as in [11], except for the expected number of secondary infections, $R$, which was updated to match the hypotheses made in [12], to ensure comparability of results; cf. Table 2. The data on early Alpha cases were retrieved from the Global Initiative on Sharing Avian Influenza Data (GISAID) [36], available at https://doi.org/10.55876/gis8.230104xg (see S1 Text). We use the data on the sequences submitted to GISAID up to November 30, 2020, and used only the samples collected up to November 11, 2020. This choice was done to overcome reporting delays, and to exclude sequences analysed and added retrospectively once the growth of the variant had been identified; cf. Fig D in S1 Text.

**COVID-19 in Wuhan.** The second application concerns the early COVID-19 cases reported in Wuhan, China. Here, we define a case as a confirmed COVID-19 infection, in many instances determined retrospectively [28]. We use the dataset considered in [1] (personal communication), comprising 3 072 COVID-19 cases with symptoms onset between December 10, 2019 and January 19, 2020, the day before the first public statement on human-to-human transmission [22]; cf. Fig G in S1 Text. The input parameters are summarized in Table 2.

## Supporting information

**S1 Text. Appendix. Fig A. Reanalysis of Roberts et al. (2021) [4] with updated datasets**. The original analysis was done with Huang et al.'s dataset [5]. We re-ran the analysis on updated case datasets, using the same $N = 10$ number of case-days as in the original analysis. **Fig B. Distribution of the difference (in days) between the estimated date of the first infection and the actual date of the first infection in simulated data**, for different values of $N$, the number of cases considered for the evaluation. The vertical lines represent the median (full line) and the mean (dashed line) of the distributions. The different simulated datasets are gathered together. **Fig C. Estimated dates of first infection, for each of the 100 simulated datasets**. The dates of first infection in the simulated datasets are set at 0. The squares show the dates at which there were $N = 1, 10, 1000$ cases in the simulated dataset, while the diamond, cross and bars show the mean, median and 95% interval of estimated dates of the first infection corresponding to each $N$, using the simulated dataset as source data. A perfect estimation

lands on 0. **Fig D. Alpha sequences in the UK**. We apply our model to 406 sequenced samples (blue) collected up to November 11 (light-blue highlight) and reported by November 30, 2020 (date of submission to GISAID [36]). We do not consider samples collected between November 12 and November 30 to avoid the effects of reporting delays in the last days. For comparison, here we present the sequences reported after November 30, and up to December 15 (white), corresponding to a total of 455 sequenced samples. **Fig E. Cumulative cases of Alpha variant in the UK**. Accepted simulations (gray) and observed data (black). **Fig F. Estimates of the emergence date using information on the first case only**. Distributions of the emergence date for the Alpha variant in the UK, estimated using data on the first observed Alpha case (i.e., September 20, 2020) only, using our model (violet, upper middle) and the model previously developed by (Czuppon et al., 2021) as published (yellow, bottom) and updating the parameters to match ours (red, lower middle; cf. the Methods section of the main text). We also plot the results from running our model on data of samples collected up to November 11, 2020 (i.e., our main results; blue, top), for comparison. **Fig G. Early COVID-19 cases in Wuhan**. We apply our model to 3072 COVID-19 reported cases with symptoms onset by January 19, 2020 (blue). On January 20, the first public declaration of human-to-human transmission of the virus was made. Soon after that, the lockdown intervention was deployed nationally, along with testing, which explains the change in epidemic dynamics observed after that date (white). **Fig H. Cumulative cases of COVID-19 in Wuhan**. Accepted simulations (gray) and observed data (black). **Fig I. Estimates of the emergence date using truncated or outdated case datasets**. Distributions of the emergence date for the COVID-19 epidemic in Wuhan, estimated using data on cases with symptoms onset by December 31, 2019 (violet, middle) and by January 19, 2020 (i.e., our main results; blue, top), as well as an outdated, later corrected dataset (red, bottom). The first cases reported in the datasets are depicted by dashed lines colored accordingly. **Table A. Model calibration**. Our results are obtained from a set of 5 000 simulations selected by the model calibration. Here, we summarize some metrics obtained by calibrating our model for the two applications, and show the observed data for comparison. The values and ranges of estimates correspond to medians and 95% interpercentile ranges. **Table B. Estimates for the emergence date**. Estimated dates of emergence. Median values and central 95% interpercentile ranges (values between 2.5$^{th}$ and the 97.5$^{th}$ percentiles) are shown. Abbreviations: tMRCA = time of most recent ancestor. **Table C. Sensitivity analyses**. Estimated dates of emergence, obtained by varying the key epidemiological parameters (reproduction number, over-spreading and detection) as well as the parameters regarding the model calibration (i.e., the tolerances for simulation selection; see details in the Methods section of the main text). Median values and 95% interpercentile ranges (values between 2.5$^{th}$ and the 97.5$^{th}$ percentiles) are shown. Baseline values are marked in boldface. **Table D. Impact of using different datasets**. Results obtained using different COVID-19 cases datasets. The estimated time elapsed between the first infection to the $N^{th}$ observed case yields the estimated date of outbreak emergence. In addition, we estimate the epidemic size at the date of detection of the $N^{th}$ case. The proportions of detected infections are retrieved for comparison with the input epidemic parameters (cf. Table 2 of the main text). Median and 95% interpercentile ranges (i.e., values between 2.5$^{th}$ and the 97.5$^{th}$ percentiles) are shown, unless stated otherwise. (PDF)

## Acknowledgments

We thank Verity Hill [12] and Jonathan Pekar [1] for sharing their data and results for comparison with ours. We also gratefully acknowledge all data contributors, i.e., the Authors and their Originating laboratories responsible for obtaining the specimens, and their Submitting

laboratories for generating the genetic sequence and metadata and sharing via the GISAID Initiative, on which this research is based. We thank four anonymous reviewers for comments on the manuscript.

## Author Contributions

**Conceptualization:** Peter Czuppon, François Blanquart, Florence Débarre.

**Data curation:** Florence Débarre.

**Formal analysis:** Sofía Jijón, Peter Czuppon, François Blanquart, Florence Débarre.

**Funding acquisition:** Florence Débarre.

**Investigation:** Sofía Jijón, Peter Czuppon, François Blanquart, Florence Débarre.

**Methodology:** Sofía Jijón, Peter Czuppon, François Blanquart, Florence Débarre.

**Project administration:** Florence Débarre.

**Resources:** Florence Débarre.

**Software:** Sofía Jijón, Peter Czuppon.

**Supervision:** François Blanquart, Florence Débarre.

**Validation:** Peter Czuppon.

**Visualization:** Sofía Jijón, Florence Débarre.

**Writing – original draft:** Sofía Jijón.

**Writing – review & editing:** Sofía Jijón, Peter Czuppon, François Blanquart, Florence Débarre.

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
