## [Decision Letter · Decision Letter 0]

18 May 2023

Dear Dr Jijón,

Thank you very much for submitting your manuscript "Using early detection data to estimate the date of emergence of an epidemic outbreak" for consideration at PLOS Computational Biology.

As with all papers reviewed by the journal, your manuscript was reviewed by members of the editorial board and by several independent reviewers. In light of the reviews (below this email), we would like to invite the resubmission of a significantly-revised version that takes into account the reviewers' comments.

I agree with the reviewers that this is an interesting paper, but adding simulation studies and a thorough investigation of how violations of assumptions around things like sampling impact the performance of the methods is needed.

We cannot make any decision about publication until we have seen the revised manuscript and your response to the reviewers' comments. Your revised manuscript is also likely to be sent to reviewers for further evaluation.

Sincerely,

Samuel V. Scarpino

Academic Editor

PLOS Computational Biology

Thomas Leitner

Section Editor

PLOS Computational Biology

Reviewer's Responses to Questions

**Comments to the Authors:**

Reviewer #1: Review uploaded as attachment.

Reviewer #2: See the review attached

Reviewer #3: This is a well-written and interesting study on how to infer the date of first infection of an epidemic using a branching process modelling approach. I have only minor comments.

1. l. 277. I am not sure whether the model takes into account super-spreading events when the generation times of infections are independently distributed (l. 284). Rather than superspreading events, wouldn't it be superspreading individuals that are taken into account?

2. l. 306-321 This is the only part of the manuscript that I have had a hard time understanding. The notion that "all simulations can actually be retained and shifted" (l. 316) was lost on me, especially since afterwards I read that in fact there is only a set of "accepted" simulations (l. 321). A more mathematical or graphical explanation of this procedure would help as there does not seem to be any mention of it in the pseudo code.

Typos:

l. 252 "constraint" -> constrained

l. 332 the sentence seems to be grammatically wrong.

Reviewer #4: This paper presents a method for estimating the origin date of an epidemic based only on a time series of cases, contingent on some fairly strong assumptions (exponential growth, all observed cases from the same spillover/introduction, constant parameters). The model is a straightforward extension of one presented in Czuppon et al (the offspring distribution is now negative binomial and the time series is used to reject simulations that deviate too far from it). These extensions make sense, but on the one example where the two methods are compared to each other the results are only slightly different, which makes it difficult to judge the impact of the extensions.

I think the model has a place, for example obtaining quick estimates of the time since a virus spillover or a new variant introduction in scenarios where sequencing data are not available or informative. It should be kept in mind that this is not a thorough investigation into the origin of the COVID-19 pandemic, nor is it a method that more accurately estimates the origin date of an epidemic than existing methods (at least that is not what is shown here). On the two examples shown here this model performs similarly to previous methods, however it is straightforward to come up with an example where previous methods that also rely on genetic data would outperform this model, such as an epidemic from multiple zoonotic spillovers, with each spillover resulting in a substantial number of cases (see Figure 1d in https://www.nature.com/articles/s41564-018-0296-2 for a hypothetical example). This is not a long paper, but it still has a fair amount of unnecessary repetition (e.g. most of section 2.1 is repeated elsewhere and I think Tables 1 and S1 could be merged).

# Major comments

1. Using the model to estimate something that is not conclusively known is not model validation. The only thing shown here is that the model recovers similar estimates to previous models, with slightly tighter credible intervals. This does not prove that this model's estimates are better in any way. It also does not prove that the model was implemented correctly or that the model is an unbiased estimator for the date of the first case, when all assumptions are met. A small simulation study would be a good way to investigate these questions. It could also be used to shed light on robustness to model violations.

2. There is a small discussion on NPIs in Wuhan affecting estimates, but nothing about England during the emergence of Alpha. Various changes in NPIs occurred in England during October 2020 and the country went into a national lockdown on November 5th. Although the NPIs in place didn't stop Alpha from spreading it likely did have some effect on the reproductive number.

3. Could the authors explain how they arrived at the fixed parameter values used here? Although the authors provide references, I think these parameter estimates are contentious and it's not clear to me why those references were chosen. In particular, the sampling probability seems like a difficult parameter to fix a priori. I think it's crucial for parameterising the model and I don't have any intuition about how one would estimate it at the start of an outbreak.

4. I wonder how much the method is affected by violations to the sampling assumptions. In the model a constant sampling probability is assumed. However, we know that cases are much more likely to be ascertained on weekdays than weekends. When sequences are used as a time-series of cases fluctuations in sampling are more extreme, since sequence surveillance is not as uniform as case surveillance and it is common for the weekly sequencing capacity to be exceeded during phases of exponential growth (but this was not the case in the Alpha dataset used here). I would expect that violations to this assumption (e.g. no samples for a week or a few days with outsized contributions) could result in unnecessarily rejecting a large number of simulations and arriving at biased estimates. Moreover, if the tolerance parameters are too strict I think the model could recover fluctuations in sampling over time. Would this be the desired outcome?

5. Could the authors explain in more detail why their algorithm is not ABC? If the approximate posteriors obtained are identical I would argue that it is just a clever algorithm for performing ABC inference. I think the authors could also be more rigorous in calling the estimated distributions approximate posteriors and not just posterior distributions.

# Minor comments

6. I'm pretty sure EU1 is not B.1.1778. If memory serves it's just B.1.177.

7. There may be only 406 Alpha genomes that were uploaded to GISAID by Nov 30th 2020, but there are thousands of known Alpha cases until then thanks to SGTF. See https://www.science.org/doi/full/10.1126/science.abj0113 and https://www.nature.com/articles/s41586-021-03470-x%22%22. These cases could serve as an extra robustness analysis.

8. Why do the authors only show the 95% IqR limits for the estimates from the new model in the figures?

9. It doesn't look like the first case arrow in Figure 2 is lined up with the first case.

10. Table S2: The Alpha TMRCA estimate is also a phylodynamic model. It relies on the coalescent model.

11. The title of Figure S6 is misleading. Are the authors suggesting that the data in Pekar et al 2022 are outdated?

12. Based on the text in the methods I think the \\kappa_t and \\kappa_tau parameters in Table 2 should probably be \\omega_t and \\omega_tau

13. Line 332: There's a missing word or two in the sentence.

14. On all the figures the date axis ticks are not uniformly spaced.

15. This is just an observation and not about this paper. It seems like the authors had to reach out to the authors of both Hill et al 2022 and Pekar et al 2022 to get the data they used in their study. It's great that they provided the data, but as both papers have been published for a while it is a little disappointing to see that the data (which are not sensitive) are still not publicly available!

**Have the authors made all data and (if applicable) computational code underlying the findings in their manuscript fully available?**

Reviewer #1: Yes

Reviewer #2: None

Reviewer #3: Yes

Reviewer #4: Yes

PLOS authors have the option to publish the peer review history of their article (what does this mean?). If published, this will include your full peer review and any attached files.

Reviewer #1: No

Reviewer #2: No

Reviewer #3: No

Reviewer #4: No
---

## [Decision Letter · Decision Letter 1]

11 Jan 2024

Dear Dr Jijón,

Thank you very much for submitting your manuscript "Using early detection data to estimate the date of emergence of an epidemic outbreak" for consideration at PLOS Computational Biology. As with all papers reviewed by the journal, your manuscript was reviewed by members of the editorial board and by several independent reviewers. The reviewers appreciated the attention to an important topic. Based on the reviews, we are likely to accept this manuscript for publication, providing that you modify the manuscript according to the review recommendations.

Sincerely,

Samuel V. Scarpino

Academic Editor

PLOS Computational Biology

Thomas Leitner

Section Editor

PLOS Computational Biology

Reviewer's Responses to Questions

**Comments to the Authors:**

Reviewer #1: I thank the authors for fully addressing my comments.

A few very minor final comments/suggestions:

Line 108: should “infected” be “detected”? (And similarly elsewhere – lines 159 and 185, and in table 1, penultimate row, first column, and perhaps other places).

Lines 195-198: I wonder whether (at least to some extent) the higher skewness could be an artefact of the fact that with a mean first infection date closer to the first reported case date, there are fewer possible first infection date values above the mean. If you’re not sure of this, I think this sentence could just be deleted.

Line 259: should “in” be “on”?

Figure S7: a few things don’t seem quite right here: (i) I think the positions of the violins (bottom/middle/top) are stated incorrectly in the caption; (ii) the violin for the model from Czuppon et al. (2021) here seems to be different from the corresponding one in Figure 2, but I can’t see why it should be; and (iii) the current figure does not seem to support the claim in line 164. Please check and update accordingly.

Algorithm 1: I think lines 37 and 39 need to be updated to reflect the updated conditions.

Reviewer #2: See the review attached.

Reviewer #4: I think the authors have sufficiently addressed my concerns. Thank you for better explaining the difference between their method and ABC. I would have liked to see a more detailed simulation study (how well can the method estimate other parameters and sensitivity to other parameters, especially the sampling probability), but given that the only real parameter of interest here is the date of the first infection, I think this is sufficient.

I have three small notes for the authors:

- Consider using another term instead of interquantile range (perhaps percentile?). I don't see interquantile used very often and the abbreviation IQR is used for interquartile range (so a 95% IQR doesn't make sense at first read).

- I still find it odd that the first case doesn't show up in Fig 3A. Is it visible when zoomed into a vector graphics version of the figure?

- A simple hack to include genomic data on top of case data is to use the tMRCA as a rejection condition. By the assumptions of the model the first infection event in the current transmission chain is at least as old as the tMRCA. Thus, any simulation with a shorter time between the first infection and the Nth case than between the tMRCA and the Nth case should be rejected.

**Have the authors made all data and (if applicable) computational code underlying the findings in their manuscript fully available?**

Reviewer #1: Yes

Reviewer #2: Yes

Reviewer #4: Yes

PLOS authors have the option to publish the peer review history of their article (what does this mean?). If published, this will include your full peer review and any attached files.

Reviewer #1: No

Reviewer #2: No

Reviewer #4: No

Figure Files:

Data Requirements:

Reproducibility:

References:

---

## [Editor Report · Decision Letter 2]

20 Feb 2024

Dear Dr Jijón,

We are pleased to inform you that your manuscript 'Using early detection data to estimate the date of emergence of an epidemic outbreak' has been provisionally accepted for publication in PLOS Computational Biology.

Best regards,

Samuel V. Scarpino

Academic Editor

PLOS Computational Biology

Thomas Leitner

Section Editor

PLOS Computational Biology

---

## [Editor Report · Acceptance letter]

5 Mar 2024

PCOMPBIOL-D-23-00121R2 

Using early detection data to estimate the date of emergence of an epidemic outbreak

Dear Dr Jijón,

I am pleased to inform you that your manuscript has been formally accepted for publication in PLOS Computational Biology. Your manuscript is now with our production department and you will be notified of the publication date in due course.

With kind regards,

Anita Estes
